# A Review of Artificial Neural Network Models Applied to Predict Indoor Air Quality in Schools

**DOI:** 10.3390/ijerph20156441

**Published:** 2023-07-25

**Authors:** Jierui Dong, Nigel Goodman, Priyadarsini Rajagopalan

**Affiliations:** 1Sustainable Building Innovation Lab., School of Property, Construction and Project Management, RMIT University, Melbourne, VIC 3000, Australia; nigel.goodman@anu.edu.au (N.G.); priyadarsini.rajagopalan@rmit.edu.au (P.R.); 2HEAL National Research Network, Canberra, ACT 2601, Australia; 3National Centre for Epidemiology and Population Health, The Australian National University, Canberra, ACT 2601, Australia

**Keywords:** indoor air quality, schools environment, neural network algorithms, artificial neural networks, predictive model

## Abstract

Background: Indoor air quality (IAQ) in schools can affect the performance and health of occupants, especially young children. Increased public attention on IAQ during the COVID-19 pandemic and bushfires have boosted the development and application of data-driven models, such as artificial neural networks (ANNs) that can be used to predict levels of pollutants and indoor exposures. Methods: This review summarises the types and sources of indoor air pollutants (IAP) and the indicators of IAQ. This is followed by a systematic evaluation of ANNs as predictive models of IAQ in schools, including predictive neural network algorithms and modelling processes. The methods for article selection and inclusion followed a systematic, four-step process: identification, screening, eligibility, and inclusion. Results: After screening and selection, nine predictive papers were included in this review. Traditional ANNs were used most frequently, while recurrent neural networks (RNNs) models analysed time-series issues such as IAQ better. Meanwhile, current prediction research mainly focused on using indoor PM_2.5_ and CO_2_ concentrations as output variables in schools and did not cover common air pollutants. Although studies have highlighted the impact of school building parameters and occupancy parameters on IAQ, it is difficult to incorporate them in predictive models. Conclusions: This review presents the current state of IAQ predictive models and identifies the limitations and future research directions for schools.

## 1. Introduction

### 1.1. Background

Air pollution exacerbated by climate change is a major challenge to public health. For example, heatwaves and wildfires increase human exposure to air pollution and unpredictable health risks [1,2,3,4]. Current research has shown that air pollution is associated with respiratory diseases, such as asthma and allergies [5], inflammatory bowel diseases [6], and lung cancer [7]. The projected impact of wildfires includes a doubling of heat-related deaths, increased hospitalizations for asthma, pneumonia, and cardiovascular effects, and increased mortality and hospitalizations associated with ozone [8]. During wildfires, the combined effect of high temperature and air pollution is more than what would be expected from the sum of their individual effects [9]. Parry et al. [10] demonstrated that air pollution modifies the association between heatwaves and hospital admissions for cardiovascular disease. Meanwhile, the health impacts of climate change are still likely to be underestimated. For example, climate change and global warming modify the availability and distribution of plant-derived allergens such as more intense and prolonged pollen seasons, increasing in the severity and alteration of the seasonality of symptoms of allergic rhinitis and asthma [11].

As a vulnerable population, children are more susceptible to air pollution. Children will likely experience much higher personal exposure, even at the same pollutant concentrations as adults [12], because infants and children inhale and retain larger amounts of air pollution per unit of body weight than adults [13]. Therefore, the National Ambient Air Quality Standards set by the United States Environmental Protection Agency (EPA) set stricter limits on the concentration of carbon monoxide (CO), nitrogen oxides (NO_x_), particle matter (PM), and sulfur dioxide (SO_2_) for children [14]. However, the World Health Organization (WHO) found that there are still excess deaths at low concentrations of these pollutants [15,16]. Currie et al. [17] pointed out that lowering the thresholds for acceptable air pollution levels may be a prudent and necessary step toward improving population health, especially among children.

People spend 90% of their time indoors [18], and children spend up to 1075 hours in classrooms each year [19]. Indoor air quality (IAQ) in schools can be further complicated by factors including location, occupancy rates, space allocation, teaching activities, the presence of mould, and ventilation [20,21,22,23,24].Classrooms could be potential important sources of exposure to bioaerosols such as allergens, fungi species, bacteria and viruses especially influenza and SARS-CoV-2. Low levels of exposure might still cause adverse health outcomes [25,26] and spread of epidemic diseases. Another concern is that classrooms are often inadequately ventilated [27,28], which results in health risks and reduces academic performance [29]. Airborne disease transmission of respiratory viruses has drawn significant attention in indoor bioaerosol research since the COVID-19 outbreak. Site-specific strategies are necessary due to the characteristics of school buildings [30], while also taking into account the socio-economic consequences of health and performance effects on children.

### 1.2. Sources of Indoor Air Pollutants (IAP) in Schools

Similar to other building types (e.g., residential, commercial), there are many different sources of IAP in schools, including CO, black carbon (BC), NO_x_, SO_2_, PM, ozone (O_3_), and volatile organic compounds (VOCs) [31,32]. These can be derived from ambient air pollution, including traffic [33,34,35], factory emissions [36], wood smoke, or from indoor sources, such as furniture [37,38], building materials [39], fragrance products [40,41], and human activities. IAP in schools is also influenced by teaching activities and equipment [42]. Concentrations can depend on the location, local climate, and nearby sources, as well as potential sinks, including the proximity and extent of urban greenspace [43,44]. Compared to other regions of the world, the concentrations of key pollutants (e.g., PM_2.5_) in Australian cities are generally lower although occasional exceedances in ozone and PM_2.5_ can occur during extreme pollution events, such as bushfires, dust storms, or heatwaves [45]. In Australia, the 24-hour threshold of PM_2.5_ defined by the National Environment Protection Measure (Air NEPM) [46] is 25 μg/m^3^, the same as WHO standards. However, during bushfires, the concentrations of PM_2.5_ measured in Sydney exceeded 100 μg/m^3^ and could reach 500 μg/m^3^, which is four times higher than the guideline [47]. Therefore, IAP caused by extreme climate events needs to be treated with extra attention.

### 1.3. Influencing Factors

Many factors can affect IAQ, with complex connections between them. These factors can be divided into the physical conditions within a building and its operation. Physical conditions include factors such as materials, room size, window-to-wall ratio, and air tightness of the building envelope, while the operation mainly refers to the ventilation system, such as ventilation method (mechanical and natural ventilation), supply air temperature and humidity, air exchange rate, and human activities [39,48,49]. IAQ in schools is specifically influenced by occupant density and activities. First, as school schedules and students gather, CO_2_ concentrations subsequently increase. Second, the frequent movement of students in and out of classrooms leads to the re-suspension of airborne particles [20], which makes IAQ more unpredictable. Third, school activities and materials may produce uncommon pollutants, such as chemistry experiments, art tools, chalk dust, and printing [44,50,51].

The effects of these factors are complicated since they interact with each other and have a synergistic effect on IAQ [52]. For example, based on pollutant monitoring data from major EU cities, Spiru et al. [53] found that, compared to naturally ventilated buildings, mechanically ventilated buildings reduce indoor PM_2.5_ concentrations but increase the concentrations of indoor NO_2_. Meanwhile, factors such as climate, cultural context, building characteristics and interior design features influence user behaviours [54,55], which also affect IAQ.

### 1.4. Evaluation of IAQ

Existing research typically uses two types of indicators: indicators based on one or more pollutant concentrations, such as a combination of several different pollutant concentrations, including PM_2.5_, VOCs and NO_2_, and indicators of IAP exposure levels or health effects.

#### 1.4.1. Indicators Based on Pollutant Concentrations

One of the most popular indicators is carbon dioxide (CO_2_) concentration [56,57]. Research on schools showed that CO_2_ concentration is associated with sick building syndrome (SBS) [58] and affects academic performance [59]. However, there is very limited evidence that CO_2_ levels below 5000 ppm influence health symptoms, and the results on the effects of moderate CO_2_ levels on human cognitive performance are not consistent [60]. These findings are likely due to inadequate ventilation elevating the concentrations of other harmful pollutants (e.g., VOCs) along with CO_2_ levels [61]. Indoor CO_2_ can be much higher than outdoor mainly due to production by humans, especially in crowded buildings, such as schools. Although the total CO_2_ generation rate of adults is approximately 18%–54% higher than that of children, the difference between adults and children was not significant at low activity levels, such as sedentary [62]. Meanwhile, many studies used the concentrations of PM_2.5_, PM_10_, NO_2_, O_3_, CO, VOCs, and fungi to describe IAQ, both in schools and other buildings [21,63,64,65,66,67]. Considering the relationship between pollutant emissions and environmental parameters such as temperature and humidity [68,69], some research added those environmental parameters to IAQ indicators [70,71,72,73,74], while others presented them as contextual factors. However, there is a lack of clear consensus on which or what combination of pollutant parameters should be used to describe IAQ [75].

#### 1.4.2. Indicators of Health Effects and Exposure Levels

Many studies have further focused on the health risks of people exposed to IAP. Carslaw et al. [76] defined a novel method named secondary product creation potential (SPCP) for ranking indoor VOCs in terms of potentially harmful product formation. Although this sequencing can vary with room characteristics and ventilation type, it shows great potential for identifying the potential hazards of VOCs. Baloch et al. [77] used the total VOCs score as the sum of five VOCs (formaldehyde, benzene, naphthalene, limonene and tetrachloroethylene) to elevate the multi-exposure in the classroom. However, this indicator did not cover other common IAPs, such as PM_2.5_. Taking into account exposure levels, thresholds, and cumulative exposures, López et al. [78] calculated and compared 10 different CO_2_-based IAQ indicators, including the mean CO_2_ concentration, the mean concentration above a threshold value, the percentage of time spent in a concentration range, and cumulative exposures greater than a threshold value. However, the exposure limit values (ELVs) for IAP defined by WHO, European Commission, and authorities in 11 countries varied widely, even by order of magnitude for the same averaging exposure periods [75]. 

For health risks, Logue et al. [79] combined available disease incidence and disease impact models for specific pollutant-disease combinations with data on measured concentrations to estimate the disability-adjusted life-years (DALYs) lost due to inhalation of indoor pollutants. DALYs are a composite indicator of disease and injury burden, capturing both premature mortality and the prevalence and severity of ill health [80]. Furthermore, Cony Renaud Salis et al. [75] proposed three IAQ indicators (Short-Term Exposure Limit, Long-Term Exposure Limit, and IAQ-DALYs) to calculate the acute and chronic effects separately. Another health indicator is peak expiratory flow rates (PEFR), which is commonly used for ambulatory evaluation of asthma [81], as well as air pollutant concentration [82] and exposure levels [83]. For the health risk analysis of IAP exposure, characteristics of the indoor environment and pollutants need to be included, as well as vulnerability, which includes sensitivity (personal characteristics such as age, gender, race, socioeconomic status, pre-existing disease) and adaptive capacity (individual or community-based coping and adapting mechanisms) [84]. Schibuola et al. [85] specifically examined the exposure of children in school by hazard index (HI) and cancer risk (CR), which derived from the US EPA [86]. However, they only calculated the risk from CO_2_ and PM as the measured concentrations of CO, ozone, and VOCs were below the limits. It probably underestimates the health risk in school, because IAPs such as benzene, polycyclic aromatic hydrocarbons, and ozone, do not have a safe exposure threshold [16,87].

### 1.5. Machine Learning Predictive Models

Predictive models can help and account for future uncertainties, especially for predicting air pollution patterns and health effects in the context of climate change. Predictive research related to indoor environments involves several scientific fields. Predictive models can be divided into physics-related building simulation (Computational Fluid Dynamics, CFD) and mathematics-related machine learning algorithms [88]. CFD can provide detailed predictions of airflow [89], heat transfer [90] and contaminant transportation and distribution [91]. In addition, CFD can be used to study the spread and propagation of viruses in different scenarios [92]. However, CFD modelling requires detailed information about the indoor space to represent the location digitally using a fine grid, which means it requires substantial computational costs. In contrast, as a data-driven method, machine learning predictive models skip the time-consuming physical equations and derive associations between features from the dataset. Due to fast calculation times and high accuracy, machine learning has been used to assess the built environment, including in models that simulate building loads [93], building energy efficiency [94,95,96], occupancy rates [97], thermal comfort [98], as well as IAQ [99,100]. Wei et al. [99] concluded that the three most commonly used machine learning algorithms for predicting IAQ are Artificial Neural Networks (ANNs), Regression models, and Decision Tree-based models. They compared the effectiveness of those three algorithms simultaneously and found that ANNs were better for predicting nonlinear problems, such as IAQ. This finding has also been confirmed in other research on predicting IAQ [101,102].

ANNs is a data-driven algorithm based on the interconnected structure of neurons. It works on three layers (input layer, hidden layer, and output layer). The predicted output of the neural network is compared with the actual value. Based on the error, the weights assigned to each neuron are changed and then fed into the neural network again until the error is acceptable [103]. Even early ANNs using the back propagation (BP) algorithm with one hidden layer can fit continuous functions of arbitrary complexity [104]. Challoner et al. [105] used ANNs to predict indoor concentrations of PM_2.5_ and NO_2_ and found that the prediction of NO_2_ concentration was more accurate than PM_2.5_, suggesting that the correlation between input variables and output variables determines the performance of the models. For indicators of non-pollutant concentrations, Xie et al. [106] used ANNs to predict a health-related IAQ indicator, PIAQ, defined as the percentage of occupants in the sampling area of the office building with two or more persistent symptoms calculated for SBS. The input variables included six indoor air pollutants (CO_2_, PM_2.5_, HCHO, TVOCs, bacteria, and fungi) and three indoor comfort variables (temperature, relative humidity, and wind speed). With the development of machine learning, an increasing number of ANNs variants, such as convolutional neural networks (CNNs) [107,108] and recurrent neural networks (RNNs) [109,110], are being applied to IAQ research problems. As a data-driven approach, the effectiveness of an ANNs greatly depends on the training dataset. This paper reviews IAQ prediction research using neural network algorithms in schools, including classrooms and daycare centres where children gather. The review summarises data collection and factor selection in modelling processes and explores the status and limitations of IAQ prediction in schools. Finally, this paper discusses how predictive models can enhance IAQ in terms of smart building management and recommends potential research directions for the use of ANNs to study IAQ in schools.

## 2. Materials and Methods

This review was guided by the protocols outlined in the Preferred Reporting Items for Systematic Reviews and Meta-Analyses (PRISMA) statement [111], and included a four-step systematic data collection process (identification, screening, eligibility, and inclusion). As the evaluation was conducted by a single reviewer, we did not register our review, or specifically assess bias among the articles.

### 2.1. Identification

To carry out this review, the keywords “indoor air quality”, “indoor air pollution”, “predict* OR forecast*”, “neural network”, and “model* OR algorithm*” were used to search in Web of Science, Scopus, and PubMed.

### 2.2. Initial Screening

In this step, review articles and conference review papers were excluded. Conference proceedings were also excluded as they were on topics unrelated to the intended target of this review.

### 2.3. Eligibility

In the eligibility phase, research on IAQ of school buildings was expected to be retained. The following criteria were used to exclude studies based on their title and abstract: Studies not directly related to IAQ but examining parameters such as temperature, humidity, occupancy, phytoremediation using plants, outdoor pollution, energy optimization, medical diagnosis, and sensor design;Studies reporting data from residential buildings, commercial buildings, offices, agricultural buildings, industry, underground environments, laboratories, kitchens, and hospitals (i.e., not in schools);Studies focusing on the link between one specific pollution source, such as VOCs (e.g., from furniture emissions), radon (e.g., from rock and soils), NO_2_ (e.g., from traffic emissions), and indoor levels of pathogens and viruses (e.g., COVID-19);Studies focusing on the effect of IAQ, such as teaching performance and diseases, such as asthma.

Based on the full text, studies where buildings are represented as simple square or two-dimensional models were excluded. These models are difficult to validate and ignore the influence of building factors in IAQ problems. In addition, articles with the same models published by the same authors in multiple journals were screened, and articles focused on IAQ prediction were retained.

## 3. Results

### 3.1. Results of the Review Process

Figure 1 depicts the process and results of implementing this review. Publication dates were set before 15 May 2023, and non-English papers were excluded. The search resulted in 134 publications in Web of Science, 160 in Scopus, and 12 in PubMed. After removing duplicate papers, 191 articles remained for screening. In the initial screening, 13 articles were removed, and 178 articles remained for the eligibility phase. Based on the title and abstract, 156 articles were removed in the eligibility step. Then, based on the full text, 13 articles were removed. Finally, nine articles were selected for inclusion in this review.

### 3.2. Algorithms

Table 1 lists the details of the predictive models in the nine papers. The algorithms used in the articles are summarised, including traditional ANNs and RNNs.

#### 3.2.1. Traditional Artificial Neural Network

Traditional artificial neural networks are designed to emulate neuronal structures in the human brain. It includes an interconnected series of layers, including an input layer, a hidden layer, and an output layer. Each layer is composed of several interconnected nonlinear processing components called neurons or nodes to extract features of the dataset [120]. When input variables are received, the neurons continuously adjust the weights and biases, and iterate according to the error between output and target value until convergence [121]. The dataset is usually divided into a training set, a validation set, and a test set. The training set is for training the weights and biases, while the validation set is for determining when to stop training to avoid overfitting. The test set is used to evaluate the performance of the predictive model. In addition, ANNs predictive models can quantify the extent to which these input variables affect the output variables [122]. For IAQ research, the output is the indicator that describes IAQ (e.g., CO_2_ and PM_2.5_ concentration) and the input is the variables that affect IAQ (e.g., outdoor parameters, occupancy rates). A total of eight papers used traditional ANNs to predict IAQ, including research comparing multiple algorithms. These variables are summarised in Section 3.3.3.

#### 3.2.2. Recurrent Neural Network

As one of the variants of ANNs, the structure of RNNs is similar with ANNs, with the difference that the neurons of RNNs also receive their own state information and pass it around. In other words, RNNs neurons receive input variables at this moment and the information of previous data. Therefore, it is particularly suitable for analysing time-series relationships [123], such as NO_2_ and PM concentrations, which vary due to factors such as human behaviours and seasonal change [124].

However, RNNs with large time spans suffer from the “vanishing and exploding gradient problem” [125]. To address these issues, Hochreiter et al. [126] proposed Long Short-Term Memory (LSTM). The neurons of LSTM are more complex, containing “forget gate”, “input gates”, “output gates”, and cell state functions. The gates function as the controller of the information, i.e., the information to be added or removed is controlled through the gates. The forget gate is used for deciding what information in the cell state to keep or throw away. The input gate helps decide which values are to be updated (i.e., the information we want the network to memorise by cell states). The output gate tells about the information the hidden state will remember [117]. In research comparing neural network algorithms, LSTM achieved the best performance in predicting IAQ [127,128], yet those research were not set in schools. It can be noticed from the model structure that LSTM has more computational parameters compared to RNNs and thus has a slower learning speed [129]. Therefore, Sharma et al. [117] simplified the LSTM structure when predicting CO_2_ and PM_2.5_ by removing the forget gate to reduce the execution time. 

### 3.3. Modelling

This section reviews the modelling process of the predictive model, including data collection, factor selection, and variables (Table 2). Due to the different locations, building characteristics, and monitoring conditions, this review focuses on the common features of predictive models for schools, and the differences are also analysed.

#### 3.3.1. Data Collection

Table 1 shows that the data sources include field monitoring and simulation data. For data-driven prediction models, data from actual field sampling provides more information [48]. However, it is usually limited by building conditions and monitoring costs [114], such as monitoring over a large time span or multiple scenarios [118]. Simulation techniques, on the other hand, can obtain data from different scenarios of the same building. For example, Cho et al. [118] set up 241 scenarios of varying HVAC systems, including heating, cooling, heat recovery, recirculation, and ventilation, for a total of 200,000 samples. Moreover, in their subsequent training, the input variables used to predict IAQ included different scenarios and metabolic rates, which are difficult to obtain from field monitoring. However, the simulation models assume and ignore some parameters, such as air change rate and occupancy density, leading to the need for further validation of the effectiveness of such models [118].

The duration of monitoring varies from 7.3 hours [118] to over 1 year [112], which means the data include information on diurnal variations and seasonal changes. Since the routine in school buildings is fixed, the monitoring time or simulation time was chosen for working hours [112,115,117,118].

#### 3.3.2. Factor Selection

To improve the performance of predictive models, factor selection is often used to remove irrelevant factors or redundant factors for training datasets. Factor selection in the nine papers reviewed can be divided into: select indicators influencing IAQ, i.e., output variables, and select input variables for predictive models. In Table 2, Rastogi et al. [112] used Grey Relational Analysis for screening pollutants that have the most significant impact on IAQ. They measured six indoor parameters (temperature, humidity, CO_2_, CO, PM_2.5_, PM_10_) and screened PM_2.5_ and PM_10_ as the output of the predictive model. Other papers selected input variables, and the methods included sensitivity tests, correlation analysis, and Principal Component Analysis (PCA). For example, Sharma et al. [117] used the Pearson correlation coefficient to calculate the correlations between pollutants and environmental parameters (temperature, humidity, wind direction, etc.) and then developed models for predicting CO_2_ and PM_2.5_, respectively. Elbayoumi et al. [115] used univariate analysis and included ambient parameters, seasonal variation, number of people, and teaching activities as influences on PM concentrations. In this study, sensitivity analysis was used for prediction in different seasons because the weights of the input variables change according to the season. However, teaching activities were not included in input variables due to the limitation of data collection. Hu et al. [116] set four cases with different combinations of input parameters and found the accuracy was better using only historical concentration data. Notably, they also used sensitivity analysis to calculate the time length of the input variables and found a better accuracy using five timesteps (t, t−1…t−4, per hour). In the face of numerous influencing factors, Zhang et al. [48] used PCA to reduce the dimensionality of their dataset, which consisted of 23 factors. They found that over 90% of the variance in the data can be explained by four principal components for both PM_2.5_, PM_10_, and NO_2_, while five principal components were identified for ozone. Moreover, the error of their model with PCA is lower than the traditional predictive model, implying that the original input variables may contain noise.

#### 3.3.3. Variables

Table 2 shows the variables used for training, including output and input. Output variables are the parameters used to represent IAQ. Among the output variables reported in nine reviewed papers, PM_2.5_ concentration was used in seven sources, CO_2_ in six sources, and PM_10_ in four sources. Concentrations of VOCs, NO_2_, O_3_, and formaldehyde (HCHO) were also mentioned. Excluding simulation studies, pollutant concentrations were obtained from real-time sensor monitoring. Most models conducted stepwise prediction of pollutants concentrations [113,114,116,117,119] and used multiple variables as outputs, while Rastogi et al. [112] and Elbayoumi et al. [115] used daily average PM_2.5_ and PM_10_ as the output variables. 

The pollutants and corresponding concentrations reported as outputs in each table are summarised in Table 3. It should be noted that most of the papers, except for number 3 [114] and number 6 [48], used line graphs to represent pollutant concentrations. Therefore, the maximum and minimum values in the table are imprecise. Zhang et al. [48] monitored 11 classrooms, and the maximum and minimum values shown in the table are based on the average concentration in each classroom. During the modelling process, the concentrations of output variables were normalised. However, the model performance outside the extreme range is not validated. For example, the maximum CO_2_ concentration in [114] is 551 ppm, while in other papers, it exceeded 1000 ppm [112,116,117,119].

As mentioned earlier, numerous influencing factors lead to various input variables. In these nine papers, the input variables can be divided into four categories: indoor parameters, outdoor parameters, building characteristics, occupancy, and time (Table 2). All the papers used indoor parameters as the main input variables, while outdoor data and building characteristics are relatively scarce. For building-related factors, prediction studies face different obstacles in data collection. For example, factors in number 7 (number of fans, room size, and number of floors) are easy to obtain [117], while the natural ventilation rate in number 4 was estimated based on indoor CO_2_ concentration and number of people [115]. Other interventions that modified the indoor environment were not collected, impacting the model performance. For example, predictive model number 3, excluding the parameters of the HVAC system, resulted in large errors during air conditioning operation [114]. The simulation data compensates for this deficiency. Factors related to mechanical ventilation or air conditioning systems were only used in research number 8, which was trained by simulation data [118]. Due to the availability of data used in the simulation, number 8 was also the only study that used occupancy parameters as input variables. Although the number of people, clothing insulation, and metabolic rate were assumed, this model can be expected to use in real buildings.

### 3.4. Pollutants Patterns of Predictive Research in Schools

The trends in predicted pollutant levels may be similar within one building; however, it is difficult to make general conclusions for all studies due to the multiplicity of influencing factors, such as location, building structure, and monitoring conditions. Chen et al. [119] predicted CO_2_, TVOC, and HCHO in different rooms (classrooms, offices, and computer rooms) on the same floor and found TVOC was independent of room function, which meant TVOC data from a single sensor could be applied to other rooms. Patterns of HCHO in different rooms were not correlated; thus, the predictive models need to be trained separately. The patterns of CO_2_ collected varied widely by room type, i.e., the data of similar rooms can be referenced. CO_2_ concentrations in offices and computer rooms first increased and then decreased during working hours, while peaks in classroom CO_2_ occurred several times and at higher concentrations than in other rooms. It confirms the necessity of providing guidelines based on specific patterns of IAP in schools.

## 4. Discussion and Future

### 4.1. Limitation of IAQ Prediction in Schools

The vulnerability of children and the health risk of pollution exposure in classrooms are well recognised. However, a limited amount of research has been conducted to predict IAQ for schools, and even less published research has examined specific pollutants in classrooms. For example, further causes of dangerous exposures levels affecting students’ health are those related to the natural emission of radon gas, which typically accumulates in poorly ventilated classrooms, while chemicals substances (i.e., cyanoacrylate, lead, cadmium, nickel) might be contained in school materials [21]. Due to the lack of enforcement guidelines, the predictive models did not focus on specific pollutants in school scenarios. Therefore, the air quality guidelines in schools need to be updated and generalised to develop predictive models. 

It should be noted that the impact of extreme weather events was also not discussed as a scenario or variable in the nine papers reviewed. This means that these predictive models were primarily used under conditions of more regular climatic scenarios, and findings are probably not valid for extreme weather events, such as wildfires and dust storms. Two of the nine reviewed papers used outdoor parameters, such as temperature, pollutants concentration, and wind data from a public historical weather database [114,117]. Considering the trend of climate change, the combination of climate or weather prediction models may be more advantageous for IAQ prediction than using historical data. However, there is a research gap in how IAQ prediction models respond to climate hazards and create safe indoor environments.

### 4.2. Modelling Process

Current neural network models for predicting IAQ vary widely regarding algorithm structure and selection of variables. Most of the research has used traditional ANNs [48,112,113,114,115,117,118,119], while RNNs may be more advantageous for time-series problems [116]. The differences in school location, season, monitoring costs, and the building itself caused inconsistent results in the choice of variables in the nine papers. Most studies used PM_2.5_ and CO_2_ concentrations as output variables because existing guidelines and standards already set limits on them. A few studies included NO_2_ [48], which is mainly derived from daily traffic emissions. On the other hand, the input variables vary more because the influences of indoor pollution sources are more complex and uncertain. For example, indoor pollutant components and concentrations in schools located in high-traffic areas are significantly different from those in suburban areas, resulting in differences in both input and output variables and weights in predictive models. If input variables with low correlation are selected, it will cause data noise and decrease the performance of the model [48].

The modelling process is not yet standardised and is limited by building conditions and data availability. Frequent sampling and long-term monitoring typically mean higher costs, leading to challenges in the widespread use of IAQ predictive models. For data that are difficult to collect, such as personnel data (e.g., clothing insulation, metabolic rate), the corresponding simulation inputs can be used as the training dataset. In addition, the development of an algorithmic framework can help reduce data dependency. For instance, Tariq et al. [130] developed a framework for predicting pollutant concentrations in subway stations by combining transfer learning (TL) and neural networks. The study involved building a pre-training model with a well-measured subway station dataset and subsequently fine-tuning it with lesser data from other subway stations, ultimately providing well-performing prediction models for four subway stations. This TL framework, combined with pre-training and fine-tuning, can reduce the dependence on data collection. However, TL research applied to IAQ prediction is still in the beginning stages.

### 4.3. Performance and Aim of Predictive Models

The performance of black-box models is usually indicated by the difference between the predicted and actual values. Hu et al. [116] raised the problem of matching computational speed with dynamic control systems; however, their research still focused on the accuracy of the predictive model rather than the environmental benefits of the model. In fact, The models are rarely used in actual buildings due to the constraints of cost and building access, which results in current research being limited by monitoring costs [131] and computational costs [132]. Sharma et al. [117] built an IAQ platform (including monitoring, prediction, and control) by incorporating user ratings to verify the effectiveness of the platform. However, there is no data related to subsequent changes in pollutant concentration. How to calculate the benefits of the prediction model and how to assess the degree of air quality improvement is yet to be studied. 

Predictive models can be used for similar rooms in a building to reduce the cost of installing and operating multiple sensors [133]. However, this should be performed with caution as sensors have the capacity to report localised trends that may be underestimated in models, especially for the non-uniform air pollution caused by photocopiers or paints. Furthermore, models can be combined with dynamic control systems to improve IAQ. For example, Hu et al. [116] aimed to avoid unsatisfactory pollutant peaks by combining a predictive model and dynamic control, concluding that the key to solving the system control delay is to ensure that the delay time interval of dynamic prediction is less than the sampling interval of data collection. Warning and control signals require more investment in building management systems. Sharma et al. [117] developed an Android application called IndoAirSense (IAS) for predicting IAQ in classrooms. IAS provides real-time pollutant levels from sensors and prediction of pollutant concentrations, as well as user alerts and suggestions to improve IAQ. However, such an intelligent system that includes monitoring, prediction, and control needs to face not only the problem of prediction accuracy but also the coordination between building functions, such as ventilation system responsiveness, user activities, and outdoor air quality events. In other words, improving IAQ by predictive models requires alignment between predictive model performance and control system. 

## 5. Conclusions

Predictive models are useful in improving IAQ. In this paper, we reviewed studies on the prediction of IAQ in schools based on neural network algorithms. The conclusions of this paper are as follows: IAQ is currently described by the concentration of pollutants, exposure levels and health risks. In predictive models for schools, the concentration of PM_2.5_, CO_2_, PM_10_., VOCs, NO_2_, O_3_, and formaldehyde were used to represent IAQ. Other common pollutants, such as CO, SO_2_, and bioaerosols, were ignored. IAQ predictive models in schools have not yet covered the pollutants mentioned in the guidelines;Traditional ANNs and their variant RNNs were applied to predict IAQ. Traditional ANNs was the most used, while RNNs was more suitable for analysing time-series problems such as predicting IAQ. Algorithmic frameworks, such as TL, have the potential to reduce the data cost of predictive models;Many factors affect IAQ in schools (e.g., ambient environment, schedules, number of occupants, and teaching activities). However, field data were sparse, and occupancy parameters were only used in predictive models trained by simulation data, which requires further validation. Future research could focus on how to use both field data and simulation data to develop predictive models;Climate models and weather forecasts provide data that have a high potential for IAQ prediction, especially for long-term climate change and extreme weather events. Currently, ambient prediction results have not yet been applied to indoor prediction for schools;The concentrations of indoor pollutants may be similar in rooms with comparable functions, pollutant characteristics, and sources, which offers the possibility of reducing the frequency and number of field measurements. Further work is required to assess the correlation between pollutants and the potential for reducing monitoring costs.

## Figures and Tables

**Figure 1 ijerph-20-06441-f001:**
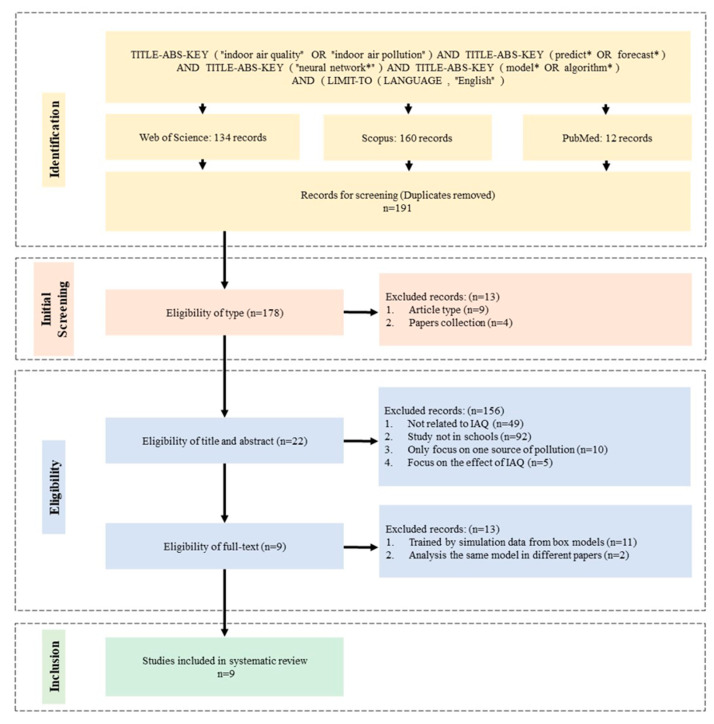
Literature review process.

**Table 1 ijerph-20-06441-t001:** Basic information on predictive models in schools.

#	Authors	Year	Location	Algorithms	Data Source	Data Duration
1	Rastogi et al. [112]	2020	University, India	Traditional ANNs	Field data	14 months, working hours
2	Kim et al. [113]	2022	Child day care centres, Korea	Traditional ANNs	Field data	1 month
3	Marzouk et al. [114]	2022	University, Egypt	Traditional ANNs	Field data	2 months
4	Elbayoumi et al. [115]	2015	4 Schools, Palestine	Traditional ANNs	Field data	8 months, working hours
5	Hu et al. [116]	2021	School, China	RNNs	Field data	27 days
6	Zhang et al. [48]	2022	University, USA	Traditional ANNs	Field data	2 weeks
7	Sharma et al. [117]	2021	University, India	Traditional ANNs, RNNs(LSTM)	Field data	32 days, working hours
8	Cho et al. [118]	2022	School, Korea	Traditional ANNs	Simulation data	7.3 hours
9	Chen et al. [119]	2018	University, Singapore	Traditional ANNs	Field data	15 days

**Table 2 ijerph-20-06441-t002:** Information of modelling process in prediction models.

#	Authors	Factor Selection	Variables					
			Output (Indoor)	Input				
				Indoor Parameters	Outdoor Parameters	Building Characteristics	Occupancy	Time
1 *	Rastogi et al. [112]	Grey Relational Analysis	PM_2.5_, PM_10_	PM_2.5_, PM_10_	-	-	-	-
2 *	Kim et al. [113]	-	CO_2_, PM_2.5_, VOCs	T, H, CO_2_, PM_2.5_, VOCs	-	-	-	Hour, day
3	Marzouk et al. [114]	-	CO_2_, T, H	AP	T, H, WS, WD	-	-	-
4	Elbayoumi et al. [115]	Univariate Analysis	PM_2.5,_ PM_2.5-10_	CO_2_, RH, CO,	PM_2.5,_ PM_2.5-10_, T, WS, RH	VR	-	-
5 *	Hu et al. [116]	Sensitivity Tests, Correlation Coefficient	CO_2_, PM_2.5_	PM_2.5_, T, RH, CO_2_, VOCs	-	-	-	-
6	Zhang et al. [48]	Principal Component Analysis	PM_2.5_, PM_10_, NO_2_, O_3_	Generate new principal component from the original data				
7 *	Sharma et al. [117]	Pearson Correlation Coefficient	CO_2_, PM_2.5_	NO_2_, CO, T, H	WS, WD	No. of fans, Room size, Floor no.	-	-
8 *	Cho et al. [118]	Stepwise Linear Regression	PMV, CO_2_, PM_2.5_, PM_10_	AP, T, RH, CO_2_, PM_2.5_, PM_10_	-	Multiple HVAC system parameters	PMV, No. of people, Clothing insulation, Metabolic rate	-
9 *	Chen et al. [119]	-	CO_2_, VOCs, HCHO	CO_2_, VOCs, HCHO	-	-	-	Hour, Day, Lag length

T: temperature; H: humidity; RH: relative humidity; VR: ventilation rate; PMV: predicted mean vote; AP: air pressure; WS: wind speed; WD: wind direction; *: use time series data.

**Table 3 ijerph-20-06441-t003:** Pollutants and corresponding concentrations reported as output variables in each of the reviewed papers.

#	Authors	Pollutants		Concentrations	
			Min	Max	Average
1 *	Rastogi et al. [112]	PM_2.5_	<50 μg/m^3^	>300 μg/m^3^	-
		PM_10_	<50 μg/m^3^	>350 μg/m^3^	-
2 *	Kim et al. [113]	CO_2_	500 ppm	1500–2000 ppm	-
		PM_2.5_	<40 μg/m^3^	160 μg/m^3^	-
		VOCs	-	16,000 μg/m^3^	-
3	Marzouk et al. [114]	CO_2_	393 ppm	551 ppm	448.7–477.9 ppm
4	Elbayoumi et al. [115]	PM_2.5_	-	-	104 ± 85 μg/m^3^
		PM_2.5–10_	-	-	350 ± 197 μg/m^3^
5 *	Hu et al. [116]	CO_2_	300 ppm	1500 ppm	-
		PM_2.5_	-	280 μg/m^3^	-
6	Zhang et al. [48]	PM_2.5_	0.63 ± 0.12 μg/m^3^	5.08 ± 3.48 μg/m^3^	-
		PM_10_	0.65 ± 0.13 μg/m^3^	5.45 ± 3.71 μg/m^3^	-
		NO_2_	7.42 ± 3.26 ppb	55.58 ± 2.19 ppb	-
		O_3_	0.03 ± 0.23 ppb	26.11 ± 3.5 ppb	-
7 *	Sharma et al. [117]	CO_2_	<200 ppm	1200 ppm	-
		PM_2.5_	<150 μg/m^3^	300 μg/m^3^	-
8 *	Cho et al. [118]	CO_2_	Simulation data		
		PM_2.5_			
		PM_10_			
9 *	Chen et al. [119]	CO_2_	500 ppm	1750 ppm	-
		VOCs	2.5 ppm	4 ppm	-
		HCHO	-	-	-

*: use time series data.

## Data Availability

No new data were created or analyzed in this study. Data sharing is not applicable to this article.

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
