# Peer review of "A Review of Artificial Neural Network Models Applied to Predict Indoor Air Quality in Schools"

_ijerph, 2023, doi:10.3390/ijerph20156441_

Round 1

Reviewer 1 Report

I thank the editor and authors for the opportunity to read and analyse this systematic review.

I am not an expert in this specific field but I have reviewed the manuscript by the methods considering it is a systematic review. For this, I have used the PRISMA and AMSTAR II checklists.

I will only list here the missing items on the checklists.

The PRISMA Abstract checklist is not followed thoroughly. I understand that this could be due to length limitations.

Even though the author's contributions are described as the journal wants them, in a systematic review the contributions of each author in each stage should be described precisely as a minimum of two persons should review each selected study. Also, the software used should be mentioned.

The risk of bias assessment is missing. If you haven't conducted a risk of bias assessment, it should be described and reasoned.

The flow diagram should be in results not in methods.

Certainty of evidence is not assessed.

Pre-registration details are missing. If you haven't pre-registered the protocol of this systematic review, it should be mentioned with reasons why it was not done.

The authors mention support from the Healthy Environments And Lives (HEAL) and RMIT Research Stipend Scholarship but no funding. This support should be described in detail if it really does not include funding.

I am aware that some of the PRISMA and Cochrane tools for systematic reviews are more designed for pure medical studies and might be hard to include in this review but at least it should be mentioned why they were not used if the PRISMA methods were followed.

Author Response

Thank you very much for your constructive comments. All the revised part is highlighted in yellow. 

Responds to Reviewer #1:

  1. The PRISMA Abstract checklist is not followed thoroughly. I understand that this could be due to length limitations.
    • Even though the author's contributions are described as the journal wants them, in a systematic review the contributions of each author in each stage should be described precisely as a minimum of two persons should review each selected study.
    • Also, the software used should be mentioned.
    • The risk of bias assessment is missing. If you haven't conducted a risk of bias assessment, it should be described and reasoned.
    • The flow diagram should be in results not in methods.
    • Certainty of evidence is not assessed.
    • Pre-registration details are missing. If you haven't pre-registered the protocol of this systematic review, it should be mentioned with reasons why it was not done.

I am aware that some of the PRISMA and Cochrane tools for systematic reviews are more designed for pure medical studies and might be hard to include in this review but at least it should be mentioned why they were not used if the PRISMA methods were followed.

Response: Thank you. We have revised the Abstract and Section 2 detailing the approach that we took for article screening. We note that our review was guided by the PRISMA statement; however, did not prescriptively follow the protocol (as the review was primarily completed by a single reviewer, was not registered, and did not specifically assess bias among the articles). We believe that we have clarified our methodological approach. We have moved Figure 1 and the relevant part to the results section.

  1. The authors mention support from the Healthy Environments And Lives (HEAL) and RMIT Research Stipend Scholarship but no funding. This support should be described in detail if it really does not include funding.

Response: Thanks for your reminding. We have revised it.

Thank you again for your suggestion. 

Reviewer 2 Report

Comments to authors are attached.

Author Response

Thank you very much for your constructive comments. All the revised part is highlighted in yellow. 

Responds to Reviewer #2:

  1. Section 1.1, Line 60, you can add a discussion of airborne viruses, especially influenza virus and SARS-Cov-2, in this line. Airborne disease transmission has drawn much attention in indoor bioaerosol research.

Response: Thanks for your suggestion. We have added it in section 1.1 as per your suggestion.

  1. Section 3.2.3, Line 46 50, can you give some statistics (such as average, median, peak concentration) of key output variables, such as PM2.5 and CO2, in the training and validation dataset this nine research? Did the pollution level have any impact on the model’s performance?

Response: Thanks for your suggestions. We have added Table 3 to describe the statistics value of output variables.  A discussion about the pollution level and models’ performance (“However, the model performance outside the extreme range is not validated.”) also added in section 3.2.3, line 50 - 61.

  1. Section 3.3, you only discussed one study, Ref [117], in this section. What about the outcome or conclusion from the other papers?

Response: Thank you for your comment. We agree that discussing only the Ref [117] is inadequate. However, only few studies gave further conclusions and we placed this as part of the discussion in section 4.3, describing the aim and use of predictive models. These nine articles did not mention the correlation between pollutant concentration levels and model performance. Given that, “outcome” as a subheading is too broad, we have changed the title of section 3.3.

Thank you again for your suggestion. 

Reviewer 3 Report

This review manuscript is well-organized, well-presented, and necessary to the specific topic about simulating indoor air quality using neuron network approach. A minor revision is suggested regarding the connection between air pollution, mostly considered for outdoor air in the first 2 paragraphs, and indoor air quality in the third paragraph (section 1.1).

Author Response

Thank you very much for your constructive comments. All the revised part is highlighted in yellow. 

Responds to Reviewer #3:

  1. A minor revision is suggested regarding the connection between air pollution, mostly considered for outdoor air in the first 2 paragraphs, and indoor air quality in the third paragraph (section 1.1).

Response: We appreciate it very much for this good suggestion, and we have Incorporated this in the beginning of the third paragraph in section 1.1 (People spend 90% of their time indoors [18] and children spend up to 1075 hours in classrooms each year [19].).

Thank you again for your suggestion. 

Round 2

Reviewer 1 Report

Dear Authors,

Thank you very much for the opportunity to review your work, and I appreciate your carefully considering my comments and suggestions.

A systematic review differs from other reviews by being transparent and replicable, reducing the amount of bias in the results. It is recommended that a systematic review needs a team of researchers to complete this task. As the lead author has done all the evaluation work in this paper, I would consider calling this paper perhaps a scoping review instead of systematic even though systematic tools were used in the process.

Otherwise, I have no other comments related to this article being published.

Author Response

Dear Reviewers,

Thank you for your feedback on previous submission. We have made the requested changes and modified the term “systematic review” as follows (highlighted in bule):

  1. The title no longer includes the word "systematic."
  2. In the Abstract-method, we have replaced "investigating" with "systematic evaluation".
  3. The heading for section 3.1 has been modified to "Results of the review process".
  4. Figure 1 has been renamed to "Literature review process".

We appreciate your input and strive to deliver high-quality content. If you have any further suggestions or concerns, please let us know.